# Shedding Light on Chemoresistance: The Perspective of Photodynamic Therapy in Cancer Management

**DOI:** 10.3390/ijms25073811

**Published:** 2024-03-29

**Authors:** Fernanda Viana Cabral, Jose Quilez Alburquerque, Harrison James Roberts, Tayyaba Hasan

**Affiliations:** 1Wellman Center for Photomedicine, Massachusetts General Hospital, Harvard Medical School, Boston, MA 02114, USA; fvianacabral@mgh.harvard.edu (F.V.C.); jquilezalburquerque@mgh.harvard.edu (J.Q.A.); hroberts4@mgh.harvard.edu (H.J.R.); 2Division of Health Sciences and Technology, Massachusetts Institute of Technology, Harvard University, Cambridge, MA 02139, USA

**Keywords:** photodynamic therapy (PDT), photodynamic priming (PDP), chemoresistance, cancer, chemotherapy, nanotechnology

## Abstract

The persistent failure of standard chemotherapy underscores the urgent need for innovative and targeted approaches in cancer treatment. Photodynamic therapy (PDT) has emerged as a promising photochemistry-based approach to address chemoresistance in cancer regimens. PDT not only induces cell death but also primes surviving cells, enhancing their susceptibility to subsequent therapies. This review explores the principles of PDT and discusses the concept of photodynamic priming (PDP), which augments the effectiveness of treatments like chemotherapy. Furthermore, the integration of nanotechnology for precise drug delivery at the right time and location and PDT optimization are examined. Ultimately, this study highlights the potential and limitations of PDT and PDP in cancer treatment paradigms, offering insights into future clinical applications.

## 1. Introduction

Cancer is a brutal disease that afflicts millions around the globe. While there are treatments which can lead to remission, they are frequently taxing on the patient or unsuccessful. There were nearly 20 million new cases of cancer and approximately 10 million deaths from cancer in 2022, according to the Global Cancer Observatory [1], and, although cancer treatments have certainly progressed, more effective treatments are desired. One of the difficulties in treating cancer is that different types of tumors manifest in the body in different ways and are attributable to different causes. Therefore, the treatment of cancers can vary largely between subtypes due to these intrinsic properties. Radiation therapy, chemotherapy, immunotherapy, and resection surgery are among the most prevalent treatments; however, each treatment has its limitations and problems. Due to this, there is a need for a therapy that can effectively kill cancerous regions while minimizing unwanted side effects. 

Chemotherapy is commonly used in the treatment of most cancers. Oftentimes, chemotherapy is used alongside other treatments, such as radiotherapy or resection surgery. The side effects are dismal and unpleasant. There are several classes of chemotherapeutics, and these groups of drugs vary widely in their mechanism, but they act systemically and indiscriminately target cancerous and non-cancerous tissues, which may collaterally kill healthy tissue [2].

Another major issue with chemotherapy is chemoresistance, which can lead to relapse, progression, and metastasis, thus ultimately resulting in poor overall survival. There are several mechanisms by which cancer can lessen the efficacy of anticancer therapeutics. For instance, the tumor microenvironment (TME) can act as a physical barrier, limiting drugs from penetrating the tumor. Additionally, heterogeneity can aid in the chemoresistance of tumors [3]. Cancerous cells can also alter the import and export of substances, for example, by altering the efflux and influx of transmembrane transporters, which limits the amount of chemotherapeutic agents present in the cells [4,5]. Once a chemotherapeutic agent is in a cancerous cell, its efficacy can be stifled in a few ways. Some forms of chemotherapy leverage damaging the DNA to kill cancerous cells, but enzymes can repair this damage, resulting in chemoresistance [6]. In summary, there are many mechanisms by which cancer can negate the effects of chemotherapeutics. 

Photodynamic therapy (PDT) could be complementary in alleviating many of the aforementioned problems. PDT involves the use of light at a specific wavelength directed at a target that has a photoactivatable compound. Upon light activation, reactive oxygen species (ROS) are generated, and cells are subsequently killed. While PDT can trigger cell death no more than 5 to 6 mm deep due to the poor depth penetration of light, a natural fallout of this therapy is the molecular and cellular properties changes in the remaining cells and in the TME. Moreover, pre-clinical studies have confirmed that PDT induces immunogenic cell death, localized inflammation, and the release of cytokines and promotes immune cell infiltration locally and remotely, potentially converting a “cold” non-immunogenic tumor into a “hot” immunogenic one [7]. These phenomena of modifying cells and, transiently, TME are referred to as “photodynamic priming” (PDP). This priming makes tumor components more susceptible to subsequent treatments, such as chemotherapy, radiation therapy, and immunotherapy [8,9]. Therefore, PDT may act as a perfect adjuvant tool to induce significant damage in the tumor tissue where the PDT dose (a product of light dose and PS concentration) is adequate, while, where the PDT is inadequate, the remaining cells and tissues are primed by increasing permeability, the destruction of the molecules responsible for drug resistance, and the infiltration of immune cells and cytokines to impact remote regions of the disease (see Figure 1). This could synergistically mitigate some of the chemoresistant mechanisms and improve the therapeutic outcomes through combination or non-overlapping mechanisms. These properties make such a therapy an ideal option for becoming a component in the armamentarium for combating several types of cancer. 

However, several limitations hinder the progress of PDT in its clinical application such as light penetration, the short lifetime of ROS, or the hypoxic environment in some types of cancer regimens. We believe that combination treatments with non-overlapping targets could work synergistically to overcome their respective drawbacks, ultimately leading to improved outcomes for cancer patients. Additionally, the mechanism of chemoresistance described below could be destroyed by PDT but only locally. This presents another limitation to using PDT to broadly overcome drug resistance. Therefore, creative timing and sequences are needed to address this issue. Herein, we sum up the most important chemoresistance mechanisms in different sections. Throughout this review, we highlight several important examples of how PDT and PDP could play a key role in addressing all these challenges. We also discuss the use of nanotechnology as an emerging approach for both PDP and light-triggered drug release in target tissues at the optimal fixed ratio. Finally, we discuss the future perspectives of PDT and PDP in terms of their clinical application.

## 2. Principles of Photodynamic Therapy and Photodynamic Priming

Photodynamic therapy is a minimally invasive treatment that involves three important elements: (i) a photoactive molecule called photosensitizer (PS), (ii) light with a specific wavelength, and (iii) molecular oxygen leading to reactive oxygen species (ROS) formation through two independent mechanisms (type I/type II reactions), which causes oxidative damage in cancer cells or pathogens (see Figure 1) [9]. Upon light excitation, one electron from the ground state is promoted to the singlet excited state (^1^PS*). The ^1^PS* can either undergo the radiative deactivation pathway through the emission of one photon, the so-called fluorescence, or can experience a fast intersystem crossing to an isoenergetic vibrational triplet state; then, internal conversion brings it into the lowest triplet excited state (^3^PS*). Despite the transition to the ground state, ^3^PS*→^0^PS is forbidden; this deactivation, called phosphorescence, can be observed due to spin–orbit coupling. In the presence of nearby oxygen molecules (^3^O_2_), the ^3^PS* can undergo a bimolecular deactivation process via electron, energy, or proton transfer. For instance, type I reactions involve an electron transfer from the biomolecules to the excited state of the PS, with the subsequent formation of a superoxide anion (O_2_^∙−^). The latter does not exhibit a significant oxidant character but could easily be transformed into a highly reactive hydroxide radical (HO^∙^). On the other hand, type II reactions involve an energy transfer from the triplet excited state of the ^3^PS to the surrounding ^3^O_2_ to produce a reactive short-lived singlet oxygen (^1^O_2_) [10]. Given ^1^O_2_’s potent oxidizing nature and significant toxicity within cells, utilizing a PS, which exhibits high ^1^O_2_ quantum yields, is preferable.

However, the generation of ^1^O_2_ is influenced not only by the characteristics of the PS structure but can also be impeded by the polarity and viscosity of the surroundings as well as the oxygen concentration in tumor regimens [11]. The ideal PS for PDT would exhibit high absorption coefficients at relatively long wavelengths, an efficient triplet state quantum yield leading to a sufficient production of ROS, long-lived emission lifetimes to enhance the probability of collisions with adjacent ^3^O_2_ molecules prior to deactivation, good photophysical stability, minimal dark toxicity, and rapid clearance [9].

Since 1993, when the first PS—Photofrin—was approved by the Food and Drug Administration (FDA) for the treatment of bladder cancer, there has been a transformative shift in clinical practices. Currently, several photosensitizers have received FDA approval for various cancer treatment regimens [12,13]. All these PSs are porphyrin, chlorin, or phthalocyanine derivatives with strong absorption in the infrared (600–700 nm) or near-infrared (700–1000 nm) region, which allows for deep tissue penetration, improving treatment outcomes. Similarly, PDP can enhance vascular and cellular permeabilization, reduce stromal components (collagen, hyaluronan, and cancer-associated fibroblasts), and activate the immune response, leading to the improvement of chemotherapy or immunotherapy’s individual therapeutic effectiveness, as shown in Figure 1 [14,15,16,17].

## 3. Photodynamic Therapy and Chemoresistance

Resistance to chemotherapeutic agents can be classified as either intrinsic (inherent to the cancer cells) or acquired (developed throughout treatment) [18,19]. Genetic mutations, the heterogeneity of tumors, and the activation of specific signaling pathways serve as intrinsic defense mechanisms of tumor cells against anticancer agents [18,19]. Alternatively, acquired mechanisms are commonly associated with changes in the levels and expression of drug targets, drug inactivation, variations in the tumor microenvironment (TME), and the mutation/activation of a proto-oncogene into an oncogene [18,19].

Nevertheless, tumor cells may exhibit both types of resistance simultaneously, including several of the hallmarks of chemoresistance, such as (1) drug efflux pumps, (2) increased DNA damage repair, (3) mutations in drug target and metabolism, (4) anti-apoptotic mechanisms, (5) epigenetic changes, (6) cancer stem cells, (7) tumor heterogeneity, and (8) tumor microenvironment (Figure 1) [20,21]. Therefore, a comprehensive understanding of the interplay between both factors and the underlying molecular mechanism of resistance is crucial for the development of new strategies and treatments to overcome this challenge [20,21]. 

PDT has emerged as an attractive candidate to overcome chemoresistance. Unlike traditional chemotherapeutic drugs which usually target specific molecules or pathways, PDT has unique multi-target features capable of simultaneously affecting multiple cellular components, such as the DNA, lipids, and proteins within cellular membranes and organelles. PDT can sensitize resistant cancer cells to anticancer agents, making them susceptible to these drugs [16]. Moreover, it can effectively target the hallmarks of chemoresistance, thereby holding promise for improved patient outcomes, as outlined below.

### 3.1. Drug Efflux Pumps

Transmembrane transporters, also known as ATP-binding cassettes (ABC), are proteins that play a key role in the translocation of several molecules across cellular membranes by utilizing the energy produced by ATP hydrolysis [20,21]. It has been reported that some of these transporters, mainly, ABCB1 (P-glycoprotein or multidrug-resistant protein, MDR1), ABCG2 (breast cancer resistance protein, BCRP), and ABCC1 (Multidrug resistance-associated protein 1 (MRP1)), are strongly related to the development of multidrug resistance in cancer. A defect or mutation in these transporters can result in their overexpression, thus leading to a decrease in drug internalization and contributing to the enhanced efflux of various anticancer agents [19].

Many PSs may act as substrates of different efflux pumps, which ultimately may influence their regulation according to cancer type, PS, and light parameters [22,23]. However, the interaction between ABC transporters and PDT can be complex. While they can be targeted by PDT because of the high levels of reactive oxygen species (ROS) produced, there is also a potential for the PSs to be pumped out of the cells [23]. 

Some studies have demonstrated that most of the PSs currently used are not substrates of ABCB1 transporters. The exception is rhodamine dyes [22]. One in vitro study showed that the affinity of tetrabomorhodamine 123 for ABCB1 conferred resistance to photosensitizer translocation, thus reducing PDT efficacy in P388 ADR (murine leukemia) cells when this transporter was overexpressed [22,24]. Another work conducted in vitro on murine colon adenocarcinoma cells demonstrated that selenorhodamines could potentially reduce the expression of ABCB1 and enable PS internalization. They also acted synergistically with doxorubicin, improving the outcome [25]. 

Unlike ABCB1, the ABCG2 transporter has been shown to serve as a substrate for many PSs, with varying results [26]. Huang et al. have demonstrated that the expression of ABCG2 was significantly downregulated when benzoporphyrin derivative (BPD) was used to prime human pancreatic cancer cells under a sublethal PDT dose (defined as photodynamic priming, PDP). Moreover, PDP was synergistic with the chemotherapeutic drug irinotecan, thus improving its uptake. By using a xenograft orthotopic animal model, they also demonstrated that PDP combined with irinotecan prevented tumor growth compared to monotherapies [26]. 

On the other hand, another study reported that the uptake of pheophorbide-a (a chlorophyll-derived PS) was decreased in HT-29 human colorectal adenocarcinoma cells that overexpressed ABCG2 in vitro and in vivo. Treatment efficacy was improved in the presence of an ABCG2 inhibitor (Ko143) [27,28].

Chemical inhibitors targeting ABC transporters and their efflux have been explored to enhance PS intracellular accumulation and reverse the ABC effects on resistant tumor cells. The presence of ABC and tyrosine kinase inhibitors (which block the efflux) increased the uptake (up to 6-fold, depending on the cell line) of different PSs, including BPD, HPPH (Photochlor), Photofrin, and 5-aminolevulinic acid (ALA), in various cancer cell lines, such as squamous cell carcinoma, fibrosarcoma, and colon carcinoma, and improved PDT efficacy [29]. 

### 3.2. Increased DNA Damage Repair

Some chemotherapeutic drugs can directly or indirectly impact the cellular DNA by damaging different stages of the cell cycle and DNA-related processes, such as replication, transcription, and repair, hence leading to cell death. However, the increased ability of cancer cells to repair the damaged DNA (induced by the DNA damage response, DDR) is a crucial factor involved in the development of chemoresistance, which results in the decreased effectiveness of anticancer drugs, increased cell proliferation, and enhanced cell survival [21,23]. 

PDT-induced oxidative stress leads to the oxidation of various biomolecules, including lipids and proteins, subsequently affecting the DNA. While it may trigger the activation of DNA repair in response to the damage, the ultimate goal is to surpass the repair capacity of cancer cells, thus producing irreversible DNA damage and resulting in cell death [23,30]. Moreover, PDT in combination with DNA repair inhibitors seems to be a promising strategy to overcome drug resistance. It has been recently reported that the combination of chlorin e6 and a DNA repair inhibitor (Olaparib) successfully prevented tumor recurrence and metastasis in vivo for 4T1 breast cancer [31]. 

### 3.3. Mutations in Drug Target

Targeted cancer therapies have emerged as potential alternative treatments over conventional chemotherapy. They are designed to target specific molecules, proteins (i.e., tyrosine kinase and nuclear receptors), or pathways associated with tumorigenesis and be efficient in cancer treatment. Particularly, the epidermal growth factor receptor (EGFR), a transmembrane protein, is an essential target for therapeutic interventions. It contains an intrinsic tyrosine kinase receptor that binds several ligands, such as the epidermal growth factor (EGF), thus initiating a downstream signaling cascade that plays a pivotal role in cellular growth and differentiation [21]. 

The overexpression of EGFR has been found in numerous types of cancers, including lung, pancreatic, ovarian, neck, brain, and breast, and, in some cases, it is correlated with aggressive phenotypes [32]. Despite the benefits of targeted therapies, cancer cells may become resistant over time owing to mutations and alterations in the expression of the targeted molecules, which can considerably impact the response to treatment. 

EGFR-PDT-targeted therapy has been an area of interest in cancer research, promoting significant antitumor effects with the development of an antibody-based treatment [32]. The recombinant human/mouse chimeric EGFR monoclonal antibody (cetuximab) that can specifically target EGFR has been explored in combination with different PSs, with encouraging results [33]. 

For example, the combination of BPD with cetuximab in xenograft in vivo models of ovarian cancer led to a significant decrease in tumor growth and a notable increase in the overall survival [33]. Another study demonstrated that hypericin-mediated PDT combined with cetuximab inhibited tumor growth in a xenograft model of bladder cancer. Additionally, this dual therapy promoted the downregulation of EGFR and increased apoptosis [34]. 

### 3.4. Anti-Apoptotic Mechanisms

Apoptosis is a natural process of controlled and programmed cell death led by intrinsic and extrinsic signaling pathways [35]. The intrinsic (mitochondrial) pathway is mediated by caspase-9, protein kinase B (Akt), and B-cell lymphoma-2 (Bcl-2) family members, while the extrinsic pathway involves death receptors and is triggered by external death ligands on the cell surface [35]. The overexpression of certain anti-apoptotic oncogenes (i.e., the Bcl-2 gene) and impaired apoptosis in various types of tumors have been associated with resistance and can contribute to tumor cell division. Thus, targeting anti-apoptotic pathways has shown to be promising in cancer treatment. Many Bcl-2 inhibitors have been demonstrated to prevent cell growth and induce apoptosis. However, mutations in the Bcl-2 gene or the upregulation of other anti-apoptotic genes to compensate can lead to chemoresistance and promote tumor growth and cell survival [35]. 

In this context, PDT can interfere with anti-apoptotic factors through multiple mechanisms, including the downregulation of anti-apoptotic proteins. PDT can trigger the photodamage of anti-apoptotic proteins, such as Bcl-2 and Bcl-xl (B-cell lymphoma-extra large), in various cancer cell lines and make them susceptible to apoptosis, as demonstrated by Kessel and colleagues [36,37]. The use of liposomal BPD-PDT been shown to lead to the downregulation of Bcl-xl and enhance the Bax/Bcl-xl ratio, promoting pro-apoptotic activity and inducing complete killing in various pancreatic cancer lines, including the gemcitabine-resistant lines [36]. The same pattern has been observed in human breast adenocarcinoma when treated with hypericin-PDT [38,39]. In a recent study, it was shown that 5-ALA-PDT is synergistic with Bcl-2 and Bcl-xl inhibitors, thereby increasing the pro-apoptotic effects in glioblastoma [40]. 

### 3.5. Epigenetic Changes

Epigenetic alterations are related to changes in the DNA either by methylation or histone modifications without necessarily affecting DNA sequencing [41]. This can influence gene expression and result in uncontrolled cell proliferation. These changes can also be associated with other resistance mechanisms, such as DNA repair, efflux pumps, and anti-apoptotic pathways, and promote drug resistance [19]. Moreover, the aberrant epigenetic silencing of tumor-suppressor genes can hinder the immune response by downregulating major histocompatibility complex I (MHC I) and tumor-associated antigens (TAA), thus making it difficult for the immune system to recognize and eliminate tumor cells [42]. 

The evaluation of PDT on epigenetics in cancer has not been widely explored and needs further investigation. However, it is known that oxidative stress is thought to influence DNA methylation and histone modifications, which can result in alterations in gene expression [43]. Therefore, PDT could indirectly affect the epigenetic process. Additionally, PDT has been used in association with histone deacetylase and DNA methyltransferase inhibitors. It has been demonstrated that the use of 5-aza-2’-deoxycytidine, a methyltransferase inhibitor, promotes antitumor responses in four types of cancer (lung carcinoma, colon carcinoma, 4T1, and EMT6 mammary carcinoma) when combined with photofrin-PDT. This effect is more pronounced in colon and EMT6 mammary carcinoma, leading to a significant increase in their survival rates [42]. Similar results have been noticed for the combination of hypericin-mediated photodynamic therapy (HY-PDT) and histone deacetylase inhibitors (HDAC) against colon cancer, resulting in the re-expression of the gene cyclin-dependent kinase, CDKN1A, which is silenced in these cancer cells [44]. 

### 3.6. Cancer Stem Cells

Cancer stem cells (CSCs) are defined as a subpopulation of cells within a tumor that can self-renew and differentiate into distinct types of cells with a high malignancy that are present in the tumor microenvironment [45]. CSCs play a significant role in tumor initiation, sustaining its growth and survival. They can also inhibit the immune response by preventing the maturation of dendritic cells [31]. Moreover, they are believed to exhibit greater resistance and possess different metabolic, genetic, and epigenetic signatures compared to non-CSCs, posing an additional challenge to overcoming resistance. Studies have demonstrated their ability to undergo differentiation into heterogeneous lineages of cells in response to chemotherapeutic drugs, hence contributing to the development of resistance and recurrence [45]. 

CSCs can also overexpress drug transporters and influence other hallmarks of cancer. Enhanced DNA repair, increased ABC transporter expression, epigenetic changes, apoptosis, and TME alterations are some of the mechanisms by which CSCs can induce drug resistance [45]. Therefore, targeting CSCs is essential for preventing metastasis, relapse, and resistance to chemotherapy.

Several studies have demonstrated the ability of PDT to effectively eradicate CSCs, promoting their differentiation and loss of stemness [46]. 5-ALA-PDT has been shown to successfully reduce the CSCs-like characteristics of head-and-neck cancer cells, making them susceptible to chemotherapy. Notably, PDT treatment has been shown to downregulate the expression of CD44 and ALDH1, which are highly tumorigenic CSCs markers [47]. Another work has demonstrated that chlorin e6 combined with hyaluronic acid and a DNA repair inhibitor (Olaparib) successfully reverse CSCs resistance in 4T1 breast cancer [31]. 

Nevertheless, some studies have indicated that CSCs can exhibit a higher resistance to PDT than non-CSCs. This has been attributed to their lower PS internalization. It has been shown that glioma CSCs, when treated by 5-ALA, showed reduced PpIX accumulation. In the same study, PS accumulation was increased by the addition of an iron chelator, thus suggesting that the mechanism involved in such resistance was independent of the ABCG2 transporter and was related to the heme metabolism [46,47]. A similar result was noticed when pancreatic cancer cells were challenged with 5-ALA-PDT. Unlike the former study, the lower PS internalization rate was due to the overexpression of ABCG2, and PS intracellular accumulation was significantly enhanced by ABCG2 inhibition [48]. 

### 3.7. Tumor Heterogeneity and Tumor Microenvironment

The heterogeneity of cancer encompasses distinct subpopulations of cells arising from genetic, epigenetic, phenotypic, and/or transcriptomic alterations in both inter- and/or intratumoral environments [49]. This diversity confers unique molecular signatures to each subpopulation. Thus, these cells may respond differently to a given treatment, thereby resulting in a poor prognosis and clinical outcome [49]. 

The TME comprises cancer and non-cancerous cells, including an extracellular matrix (ECM), blood vessels, cytokines, immune and inflammatory cells, and stroma (fibroblasts and endothelial cells) [20]. Additionally, an acidic pH and hypoxia have been identified as essential components within the TME. The interplay between these factors in the TME can substantially reduce the efficacy of treatments and play a critical role in the emergence of chemoresistance. 

Particularly, cancer-associated fibroblasts (CAFs) have gained significant attention in anticancer treatment. They can participate in the remodeling of the ECM, creating physical barriers which can limit the penetration of several components, including anticancer agents and immune cells. CAFs can also produce matrix metalloproteinase-3 (MMP-3), which contributes to epithelial–mesenchymal transition (EMT), which is associated with increased migratory and invasive properties. They have been shown to secrete metabolites (i.e., alanine and lipids) and cytokines (such as IL-6, IL-8, and TGF-β) that promote drug resistance. Additionally, CAFs can mediate immunosuppression by producing cytokines that recruit immunosuppressive cells, such as regulatory T cells (T regs) and myeloid-derived suppressor cells (MDSCs). Furthermore, the interaction of CAFs with dendric cells can also cause immunosuppressive effects by limiting the presentation of tumor-associated antigens (TAAs), thus preventing the initiation of an antitumor immune response [50]. 

PDT can be an attractive approach in cancer treatment by not only directly damaging tumor cells but also by altering the multiple components in the TME. The production of ROS can potentially impact CAFs, thus enhancing tumor permeability and increasing the susceptibility of cancer cells to different therapies. 

An elegant study designed by Saad and collaborators demonstrated that Visudyne^®^-PDT significantly reduced cellular viability in heterotypic 3D spheroids composed of human pancreatic cancer cells (Mia PaCa-2) and pancreatic CAFs (PCAFs) in a co-culture with a low PCAF percentage (25%). However, as the percentage of PCAFs increased, the efficiency of PDT was reduced, thus resulting in a higher number of viable cells [51]. 

In fact, it has been shown that CAFs can also induce resistance to PDT in cutaneous squamous cell carcinoma (cSCC). The treatment of A431 cells in CAF-derived conditioned media resulted in their resistance to methyl-aminolevulinate (MAL)-PDT. Notably, this resistance was reversed upon the addition of a TGF-β receptor inhibitor, thus suggesting the participation of this cytokine in conferring resistance to PDT. However, under the same conditions, cell proliferation was significantly reduced in the SCC13 cells. No resistance to MAL-PDT was observed against this line [52]. Therefore, the extent of PDT’s effects on CAFs may vary depending on various aspects, such as PS, light dose, and tumor type. 

## 4. Could Engineered Drug Carriers Play a Key Role in Chemoresistance?

The combination of various therapeutic approaches targeting multiple mechanisms has become an attractive strategy for managing diseases. This scenario underscores the critical importance of understanding pharmacokinetics and optimal sequential dosages. In this regard, the use of novel drug delivery vehicles that could be activated by light with a specific wavelength has unique advantages, including a superior cytotoxicity in the desired area and the ability to tune drug release while simultaneously priming the cancer microenvironment (see Figure 2) [53,54]. Herein, we will describe how light-based and engineered nanotechnology could shed light on combating multidrug resistance in cancer.

Nano-sized chemotherapeutic drugs have several attractive advantages over the conventional administration of free low-molecular-weight agents, such as their large payload capacity, toxicity reduction, specific targeting, and controlled or sustained release. Moreover, the ability to increase the half-life of the agent enhances its accumulation in the permeable vasculature of the solid tumor by the well-known enhanced permeability and retention (EPR) effect. However, this process could be hindered by different factors, such as abnormal tumor stroma, heterogeneous tumor blood flow, and high interstitial fluid pressure, thus leading to poor immune infiltration and the limited penetration of nanochemotherapeutics [55]. Therefore, localized PDP could play a key role in enhancing the efficiency of chemotherapeutics’ delivery in the tumor microenvironment (Figure 2a). For instance, M. Overchuk et al. [56] demonstrated one way to improve the accumulation and efficacy of the encapsulated chemotherapeutic drug in the tumor tissue by employing PDP prior to nanoparticles (NPs)’ injection. The authors used a bacteriochlorophyll-peptide photosensitizer that targets the prostate-specific membrane antigen. They selected a sub-therapeutic light dose that does not induce tumor necrosis, cell apoptosis, or tumor growth delay but is sufficient to induce changes in vascular tumor leakiness. To demonstrate the latter, they utilized mice bearing bilateral subcutaneous prostate adenocarcinoma tumors. One of these tumors was subjected to PDP, while the contralateral tumor was used as a dark control. After the treatment, different types of organic and inorganic NPs were intravenously administered, and their accumulation was assessed. This elegant work showed that PDP could notably enhance the loaded chemo agents’ accumulation in the treated tumor compared to the control, hence increasing the therapeutic outcomes and reducing the long-term normal toxicity. A similar approach was also assessed by C. Bhandari et al. [57] in a study in which they evaluated the delivery of a cetuximab–IRdye800 conjugate in orthotopic head-and-neck cancer after PDP. The tumor accumulation of the dye-conjugated antibody was increased by up to 138.6 ± 47.3% within 1 h of intravenous administration when the tumor had been subjected to PDP. This rapid accumulation of cetuximab in the target tissue could be a promising approach for imaged-guided surgery. On the other hand, Huang et al. [26] evaluated the PDP effect of a dual therapy by simultaneously administering two different nanoliposomes, one carrying the photoactive molecule called benzoporphyrin derivative (BPD) and the other containing the chemotherapeutic drug irinotecan (IRI) in an orthotopic pancreatic ductal adenocarcinoma (PDAC) model. As a result, PDP enhanced the tumor leakiness, leading to an 11-fold increase in the levels of intratumoral IRI for up to 72 h (IRI was cleared out of the system within 24 h without PDP). This higher accumulation was accompanied by better tumor growth inhibition, control over metastasis, and overall survival. Another alternative approach involves delivering both agents within a single nanoconstruct. For instance, G. Obaid et al. [58] described the preparation of targeted photoactivable multi-inhibitor liposomes that can simultaneously induce the light-triggered release of IRI and photodynamic action in orthotopic PDAC. A single treatment of PDT (0.25 mg/kg) and chemotherapy (9.60 mg/kg IRI; equal to a 29 mg/m^2^ human dose) doubled the overall survival. In addition, this combined treatment was demonstrated to be significantly more potent in inhibiting tumor growth compared to the use of liposomal IRI alone at the clinical dose (20 mg/kg; equivalent to a human dose of 60 mg/m^2^). Notably, they highlighted the use of PDP as an adjuvant to reduce the toxicity of chemo agents, such as IRI. Additionally, the authors demonstrated that PDT effectively decreased collagen density and collagen fiber alignment.

The use of engineered light-activated nanoparticles facilitates the overcoming of certain biological molecular factors associated with chemoresistance. Here, we discussed how ABC transporters could decrease the uptake of a chemotherapeutic drug within the cell and how PDP could modulate its action (Figure 2b). However, some photosensitizers could also be a substrate of these transmembrane efflux pumps, hindering their PDT activity. For instance, Y. Baglo et al. [59] showed that the BPD sensitizer and its nanoliposomal formulation serve as a substrate for ABCG2 and P-gp, thereby hindering the internalization of BPD in breast cancer cell lines which overexpress these transporters. To address this challenge and escape the efflux by ABC transporters, a phospholipid-conjugated BPD in its nanoliposomal formulation was developed. Notably, the study demonstrated that, with the new formulation, BPD translocation occurred independently of both ABCB1 and ABCG2, resulting in a significant enhancement in PDT efficiency in those cell lines.

A recent report showed that tumor cell lines with high levels of ABCG2 expression were more resistant to PDT when free PS chlorine e6 (Ce6) was employed. In contrast, Ce6 nanoencapsulation in pegylated positively charged nanoparticles evaded ABCG2 action [60]. Finally, C. Mao et al. [61] described another approach to overcome P-gp chemoresistance. The authors employed an anti-Pgp monoclonal antibody that is conjugated to IR700 photosensitizer. Targeted PDP and the consequent administration of Doxil^®^ led to a 5.2-fold higher accumulation in the tumor tissue.

Overcoming chemoresistance is unlikely with a single treatment, and combination-therapy-based nanotechnology may offer the most promise. A significant factor in cancer recurrence is the remarkable adaptability of cancer cells to different treatments, particularly using both inter- and intracellular signaling pathways which facilitate tumor cell survival, proliferation, and metastasis. In this regard, the upregulation of the vascular endothelial growth factor (VEGF), involved in important pro-angiogenic activity, has been observed in response to PDT or chemotherapy. To help circumvent escape from the primary therapy, S. Tangutoori et al. [62] co-delivered the photoactive molecule BPD and the anti-VEGF monoclonal antibody (bevacizumab) in a nanoliposomal formulation. In the above study, the authors reported a fast delivery of bevacizumab in the tumor tissue within 2 h of PDT. In a similar approach, B. Q. Spring et al. [63] developed PMILs that induced light-triggered cytotoxicity and the sustained release of XL184, a tyrosine kinase inhibitor. As a result, tumor regrowth was suppressed and the signaling pathways that are associated with treatment escape (such as VEGF or MET) were upregulated after PDT. In detail, the BPD photosensitizer was encapsulated in the lipid bilayer of the liposome, while the hydrophobic multi-kinase inhibitor XL184 was loaded in PLGA nanoparticles which were further encapsulated in the hydrophilic core of the PMILs. This study showed the remarkable potential of PMILs to reduce by 1000 times the dose of the chemotherapeutic drug.

## 5. Future Perspectives and Conclusions

The intricacies of cancer demand innovative drug platforms capable of targeting different mechanisms within the tumor microenvironment. We have identified several active clinical trials (see Table 1) in which PDT serves as an adjuvant treatment alongside standard chemotherapy, which is usually administered 4 weeks after PDT. This approach aims to enhance therapeutic outcomes by using photosensitizers that are already on the market, such as porfimer sodium. However, most of these clinical trials involve patients with advanced largely unresectable cancers, which decreases the success rate of PDT as measured by survival. Throughout this review, the significant potential of PDT was demonstrated in terms of its ability to re-sensitize the tumor microenvironment, reduce the metastatic burden, and address chemoresistance by increasing the therapeutic effectiveness of chemotherapy while reducing its normal toxicity in the site of action [64].

Nanotechnology-based drug delivery systems have emerged for spatiotemporally controlled therapeutic delivery to potentiate the local efficacy of a single treatment. The use of novel drug delivery systems and light as an external targeting agent offers distinct benefits, including the unified pharmacokinetics of multiple drugs at fixed ratios, minimizing off-target phototoxicity, reducing chemotherapeutics’ dosage, and, lastly, harnessing the power of one photon to trigger drug release while simultaneously priming the cancer microenvironment or image the cancer tissue using fluorescence. The pioneering advancements in nanotechnology, which enable the integration of various agents within a single vehicle, alongside the refinement of devices for effectively delivering light to tumor tissues in conjunction with standard therapies, hold tremendous promise for future cancer treatment protocols.

It is important to note that, without thoughtful protocols, PDT/PDP may face challenges in overcoming drug resistance in the long term. The efficacy of photokilling is directly influenced by the amounts of photons absorbed by the PS. Factors such as limited light penetration within the tumor tissue, the presence of interfering molecules or pigments, and the distribution of the PS within the tumor can impede this absorption. Additionally, most cancer types are hypoxic [71], thus posing an additional problem for PDT. Moreover, the short lifetime (in terms of μs) of the ROS produced upon light activation confines their action to a localized area. Therefore, although the resistance mechanism discussed throughout this text can be locally destroyed by PDT action, more clinical evidence is required to determine the optimal light dosage, assess whether the simultaneous administration of both treatments can amplify their synergy, decide whether PDT should precede chemotherapy (as illustrated in Table 1), and explore how remote PDT effects could create a window of opportunity for additional treatments to overcome resistance and reduce the metastatic tumor burden [72].

In conclusion, PDT can be a promising candidate to enhance the efficacy of various other therapeutic modalities and serve as an ideal adjuvant treatment. Nevertheless, the appropriate use of combinations like immunotherapy, chemotherapy, and biological treatments following PDT/PDP has been reported to enhance therapeutic outcomes [7,64,73,74]. Therefore, this represents a very promising approach for achieving optimal survival outcomes. In the upcoming years, we hope to witness advancements in how PDT and PDP can overcome chemoresistance and how light can manage drug release, enabling precise control over the dosage regimen at the desired site.

## Figures and Tables

**Figure 1 ijms-25-03811-f001:**
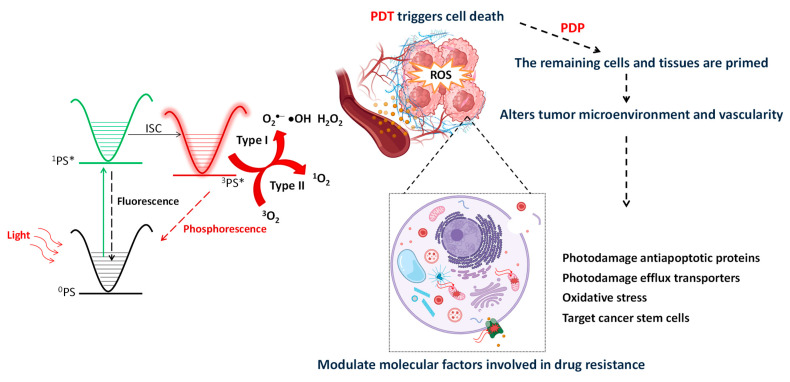
Jablonski diagram displaying the photochemical and photophysical reactions related to PDT and PDP on cancer cells and the tumor microenvironment. PDT/PDP can modulate the most important physical and molecular mechanisms of drug resistance in cancer regimens. The modification of the extracellular matrix (ECM), which serves as a physical barrier hindering the access of chemotherapeutic drugs in tumor tissues, can improve drug uptake and stimulate the immune system for improved outcomes. PDT/PDP can additionally influence key molecular factors involved in chemoresistance, including the modulation of photodamaging transmembrane efflux pumps and anti-apoptotic proteins.

**Figure 2 ijms-25-03811-f002:**
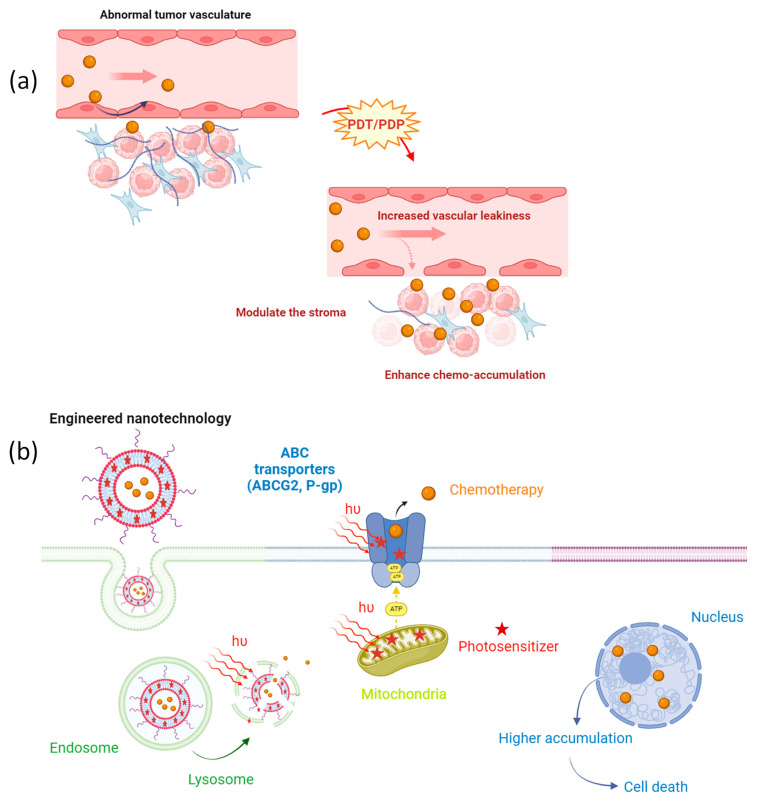
(**a**) Photodynamic action increases tumor leakiness and modulates the abnormal tumor stromal to enhance the intratumoral levels of the chemotherapeutics. (**b**) Light-activated nanoparticles act as precision tools, initiating the targeted release of chemotherapy agents (orange sphere) within the cells. When the photosensitizer (red star) located in the lipid bilayer of the liposome is activated by light (hν), we can trigger cell death but also strategically impair key cellular components such as the lysosomes, the mitochondria, or the ABC transporters. This multifaceted approach not only enhances the effectiveness of chemotherapy but also optimizes its accumulation within the targeted cells.

**Table 1 ijms-25-03811-t001:** Ongoing clinical trials to evaluate the use of PDT as an adjuvant treatment to improve chemotherapy outcomes.

Photosensitizer	Chemotherapy	Type of Cancer	Phase	Reference
Porfimer sodium	Chemotherapeutic agent, S-1	Unresectable perihilar cholangiocarcinoma	III	NCT00869635 [65]
Does not mention the photosensitizer	Epirubicin post PDT	Bladder cancer	III	NCT01675219 [66]
Porfimer sodium	Gemcitabine/cisplatin	Unresectable advanced perihilar cholangiocarcinoma	III	NCT02082522 [67]
Photosan^®^	Gemcitabine/oxaliplatin 4 weeks after PDT	Cholangiocarcinoma	II	NCT00713687 [68]
Porfimer sodium	Procarbazine 2–4 weeks after PDT	Glioma	III	NCT00003788 [69]
Fimaporfin	Gemcitabine/cisplatin chemotherapy	Cholangiocarcinoma	II	NCT04099888 [70]

## Data Availability

Not applicable.

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
