# Peer review of "Shedding Light on Chemoresistance: The Perspective of Photodynamic Therapy in Cancer Management"

_ijms, 2024, doi:10.3390/ijms25073811_

Round 1

Reviewer 1 Report

Comments and Suggestions for Authors

The manuscript makes a review of the application of photodynamic therapy (PDT) in cancer treatment and presents the most important principles of this technique. It also explores the concept of photodynamic priming (PDP) with several examples where PDP is used to enhance the efficacy of chemotherapy. The authors also discuss an additional role of incident light as an activation key for drug delivery using nanocarriers that release their contents by exposure to light.

The text is presented in a coherent and structured format, making it easy to read and understand. The core concepts are described precisely, which facilitates the understanding of the topic. In addition, the topic is likely to engage the interest of the intended readers, indicating that it is relevant in its field and could have an impact. The reference list perfectly supports the text with a considerable number of recent references in the fields of photodynamic therapy, cancer drug resistance and tumour microenvironment.

Minor corrections that should be consider:

The legend to figure 1 is too extensive. Consider shortening the legend to Figure 1 by focusing on the key points and insights presented in the figure. There is information that is superfluous because it is already included in the main text.

The concept of PDP is defined twice (in the introduction – line 63 and in section 2 – line 124). Check whether it is necessary to define the term "photodynamic process" (PDP) twice in the manuscript. Since both definitions provide unique insights, consider combining them into a single, comprehensive definition at the beginning of the document to avoid redundancy.

There is a formatting error in line 473.

Author Response

Thank you very much for your time and consideration of our manuscript. Below, you will find our responses to your comments. All changes are highlighted in yellow in the manuscript.

Reviewer 1: The manuscript makes a review of the application of photodynamic therapy (PDT) in cancer treatment and presents the most important principles of this technique. It also explores the concept of photodynamic priming (PDP) with several examples where PDP is used to enhance the efficacy of chemotherapy. The authors also discuss an additional role of incident light as an activation key for drug delivery using nanocarriers that release their contents by exposure to light.

The text is presented in a coherent and structured format, making it easy to read and understand. The core concepts are described precisely, which facilitates the understanding of the topic. In addition, the topic is likely to engage the interest of the intended readers, indicating that it is relevant in its field and could have an impact. The reference list perfectly supports the text with a considerable number of recent references in the fields of photodynamic therapy, cancer drug resistance and tumour microenvironment.

  1. Thank you for your favorable comments.

Minor corrections that should be consider:

1-The legend to figure 1 is too extensive. Consider shortening the legend to Figure 1 by focusing on the key points and insights presented in the figure. There is information that is superfluous because it is already included in the main text.

  1. Thank you for your suggestion. We have changed the captions in Figure 1.

The concept of PDP is defined twice (in the introduction – line 63 and in section

2 – line 124). Check whether it is necessary to define the term "photodynamic process" (PDP) twice in the manuscript. Since both definitions provide unique insights, consider combining them into a single, comprehensive definition at the beginning of the document to avoid redundancy.

  1. Thank you for your suggestion. We have changed this information in the text.

3-There is a formatting error in line 473.

  1. Thank you for your comment. We have addressed this issue in the text.

Reviewer 2 Report

Comments and Suggestions for Authors

In this review Cabral and collegues explore the principles of PDT and discuss also PDP. Furthermore, the integration of nanotechnologies are examined. At the end of the review the authors highlight the potential and limitations of PDT and PDP and offer insights into future clinical applications.

The review is well written and organised however some issues need to be better explained or modified before the acceptance for publication.

Line 91: authors reported that PSs are small molecules. This statement need to be better explain. PSs dimensions( in terms of molecular weight) can greatly differ from PS to PS for example ALA has a molecular weight of 131 g/mol, Foscan is 680 g/mol. If 3rd generation PSs are taken into account dimensions are higher (in line 385 you cite bacteriochlorophyll-peptide). Authors are kindly invited to better explain this.

Line 92: you cited Reactive Molecular Species but you talk only about Reactive Oxygen Species. Replace Reactive Molecular Species with Reactive Oxygen Species

Figure 1: replace this figure with Jablonski diagram that explain in the most effective way PDT

Lines 473-474: align the character with the rest of the text

General comments

Check that the acronyms appear appropriately. The full name must appear only the first time mentioned, then only the acronym must appear

Report the references in the text in square brackets as required by the journal

Increase the number of references which are currently few for a review

Author Response

Reviewer 2: In this review Cabral and collegues explore the principles of PDT and discuss also PDP. Furthermore, the integration of nanotechnologies are examined. At the end of the review the authors highlight the potential and limitations of PDT and PDP and offer insights into future clinical applications.

The review is well written and organised however some issues need to be better explained or modified before the acceptance for publication.

1 - Line 91: authors reported that PSs are small molecules. This statement need to be better explain. PSs dimensions( in terms of molecular weight) can greatly differ from PS to PS for example ALA has a molecular weight of 131 g/mol, Foscan is 680 g/mol. If 3rd generation PSs are taken into account dimensions are higher (in line 385 you cite bacteriochlorophyll-peptide). Authors are kindly invited to better explain this.

  1. Thank you for your comments. The reviewer is correct. We have changed this information in the text.

2 - Line 92: you cited Reactive Molecular Species but you talk only about Reactive Oxygen Species. Replace Reactive Molecular Species with Reactive Oxygen Species,

  1. Thank you for your suggestion. We have changed reactive molecular species to reactive oxygen species.

3 - Figure 1: replace this figure with Jablonski diagram that explain in the most effective way PDT

  1. Thank you for your comment. We have adapted the figure to clarify the photochemical and photophysical reactions of PDT on cancer cells and TME.

4 - Lines 473-474: align the character with the rest of the text

  1. Thank you for your comment. We have addressed this issue in the text.

General comments

5 - Check that the acronyms appear appropriately. The full name must appear only the first time mentioned, then only the acronym must appear.

  1. Thank you for your suggestion. We have modified this information in the manuscript.

6 -Report the references in the text in square brackets as required by the journal

  1. We have changed the references in the text.

7- Increase the number of references which are currently few for a review

  1. Thank you for your suggestion. We have included additional references in the manuscript.
